# PapB Family Regulators as Master Switches of Fimbrial Expression

**DOI:** 10.3390/microorganisms13081939

**Published:** 2025-08-20

**Authors:** Fariba Akrami, Hossein Jamali, Mansoor Kodori, Charles M. Dozois

**Affiliations:** 1Institut National de la Recherche Scientifique (INRS), Centre Armand-Frappier Santé Biotechnologie, 531 Boul des Prairies, Laval, QC H7V 1B7, Canada; fariba.akrami@inrs.ca (F.A.); hossein.jamali@inrs.ca (H.J.); 2Centre de Recherche en Infectiologie Porcine et Avicole (CRIPA), Faculté de Médecine Vétérinaire, Université de Montréal Saint-Hyacinthe, Saint-Hyacinthe, QC J2S 2M2, Canada; 3Noncommunicable Diseases Research Centre, Bam University of Medical Sciences, Bam 7661771967, Iran; mansoor92tums@yahoo.com

**Keywords:** PapB-like regulator, *Escherichia coli*, *Salmonella enterica*, fimbriae, pili, bacterial adhesion, phase variation

## Abstract

Some bacterial species within the *Enterobacteriaceae* family possess different types of fimbrial (pili) adhesins that promote adherence to cells and colonization of host tissues. One of the well-characterized fimbrial systems is the *pap* operon, which encodes P fimbriae, a key virulence factor in urinary and systemic infections. One of the key regulators of P fimbriae is the transcriptional regulator PapB which plays a pivotal role as a master switch, not only by directing phase-variable expression of its own operon but also by influencing expression of heterologous fimbrial systems. This review explores the structural organization, biogenesis, and multi-tiered regulatory control of P fimbriae, with emphasis on PapB and homologous regulatory proteins such as SfaB, FocB, PixB, and PefB. Comparative genomics and phylogenetic analyses reveal that regulators belonging to the PapB family are evolutionarily conserved across π-fimbrial systems and also regulate other types of fimbriae. These regulators respond to epigenetic changes, host-derived signals, and global transcriptional cues to control levels of production of specific fimbriae in a bacterial population to dynamically modulate bacterial adhesion in different environmental niches. Optimally, understanding these mechanisms could lead to novel approaches to perturb PapB-family proteins and abrogate production of some types of fimbriae as a targeted strategy to prevent bacterial infections dependent on adherence mediated by PapB family regulators.

## 1. Introduction

*Escherichia coli* is a facultative anaerobic bacterium that naturally colonizes the gastrointestinal tract of mammals and birds as part of the commensal microbiota. However, a subset of strains has evolved to acquire virulence traits, giving rise to various pathogenic *E. coli* pathotypes, including extraintestinal pathogenic *E. coli* (ExPEC)*,* enterotoxigenic *E. coli* (ETEC), and diffusely adherent *E. coli* (DAEC) [1,2,3]. These strains are responsible for a broad spectrum of diseases such as enteric infections, urinary tract infections (UTIs), neonatal meningitis, septicemia in humans, and colibacillosis in poultry—an economically significant condition marked by high morbidity and mortality [4,5]. Uropathogenic *E. coli* (UPEC), a major ExPEC pathotype, causes both lower and upper UTIs, while avian pathogenic *E. coli* (APEC) contributes to systemic and reproductive tract infections in birds, including peritonitis and salpingitis [2,6,7]. The virulence capability of such strains is governed by a combination of different key elements known as virulence factors [8].

Fimbrial adhesins can contribute to virulence by facilitating bacterial attachment to host cells, promoting colonization, and biofilm formation. Pathogenic *E. coli* often encode multiple types of fimbrial adhesins, allowing them to adapt to different host tissues, environmental conditions, and evade host immune responses, as the dynamic switching between adhesins can help circumvent immune detection and clearance [2,6,8,9]. Extensive research on *Escherichia coli* fimbrial adhesins has provided critical insights into their roles in host–pathogen interactions and bacterial virulence. Among these, type 1 and P fimbriae are particularly well characterized, regarding the structural organization, biogenesis, and pathogenic functions of these adhesins [8]. Type 1 fimbriae, which mediate adhesion through mannose-binding receptors, are important for the virulence of *extraintestinal* pathogenic *E. coli* (ExPEC), facilitating colonization and persistence within the host [10]. P fimbriae, first identified in uropathogenic *E. coli* (UPEC), exhibit a strong affinity for P blood group oligosaccharides, promoting adhesion to host epithelial surfaces. Due to their frequent association with *E. coli* strains isolated from pyelonephritis cases, they are also referred to as pyelonephritis-associated pili (Pap). In addition to association with UPEC, P fimbriae have been identified in some other extraintestinal pathogenic *E. coli* (ExPEC), including avian pathogenic *E. coli* (APEC) and strains linked to systemic infections in swine, highlighting their broader significance in bacterial virulence [10,11,12,13].

A critical feature of fimbrial expression across various pathogenic *Enterobacteriaceae* is its tight regulation, which enables bacterial populations to adaptively respond to changing environments during infection. This regulation involves mechanisms such as phase variation, transcriptional control, and environmental sensing, allowing bacteria to switch production of fimbriae on or off depending on the stage of infection or the specific host niche environment. Such plasticity is crucial for maximizing colonization potential, avoiding immune detection, and conserving energy. For instance, the ability to toggle between different fimbrial types or suppress fimbrial expression altogether enables bacteria to transition from adhesion to motility or from mucosal colonization to systemic dissemination [8,14]. Beyond ExPEC, the ability to regulate fimbrial expression is a fundamental strategy employed by diverse members of the *Enterobacteriaceae* family to navigate dynamic host environments. Tight regulation and phase-variable expression of multiple fimbrial systems—including but not limited to P fimbriae—enable bacteria to fine-tune adhesion, immune evasion, and tissue-specific colonization. For example, K88 (F4) fimbriae expressed by enterotoxigenic *E. coli* (ETEC) are activated in a highly controlled manner during infection of the porcine small intestine [14,15,16,17]. These regulatory systems confer phenotypic plasticity and serve as bet-hedging mechanisms, ensuring that subpopulations are primed for survival amid host immune pressures or environmental shifts. This broader perspective underscores the evolutionary significance of fimbrial regulation as a central determinant of pathogenic versatility across different bacterial species and niches [14]. This diversity of fimbrial systems reflects the evolutionary pressure on enteric pathogens to fine-tune adhesin expression for optimal fitness.

P fimbriae are defining members of the π fimbrial family [18]. The π fimbrial family encompasses a range of adhesive structures that can contribute to colonization and persistence of pathogenic *E. coli* and related enteric bacteria. P fimbriae are the prototypical representative of the π fimbrial family, alongside structurally and functionally related systems such as Pix and PL fimbriae in uropathogenic *E. coli* (UPEC), Sfp fimbriae encoded on plasmids in enterohemorrhagic *E. coli* (EHEC), and Pef fimbriae present in *Salmonella enterica*. These fimbrial systems facilitate adhesion to host tissues and are tightly regulated to adapt to changing environmental conditions and host niches [2,18,19,20,21,22]. P fimbriae are encoded by the *pap* operon, which comprises 11 genes encoding structural subunits, secretion machinery, and regulatory proteins (Figure 1). Among these, the transcriptional regulators PapB and PapI govern phase-variable expression of the *pap* genes. Notably, PapB acts as a master regulator, influencing not only its own operon but also modulating the expression of heterologous fimbrial loci.

PapB also represents a broader family of fimbrial regulators, present in different *E. coli* and *Salmonella enterica* strains [10,19,23]. Fimbrial gene expression in *E. coli* is governed by multilayered regulation operating at genetic, transcriptional, and post-transcriptional levels [24,25,26]. These mechanisms include phase variation, DNA methylation, global transcriptional regulators, and small RNAs, all of which enable reversible switching between active and repressed states. Such regulatory plasticity allows bacteria to modulate fimbrial production according to environmental conditions, such as temperature, osmolarity, nutrient status, and host signals—thereby optimizing energy expenditure and enhancing bacterial survival [24,27,28,29]. The differential regulation of multiple fimbrial types further facilitates tissue-specific adhesion and broadens the ecological adaptability of pathogenic strains [18].

This review provides an overview of the regulatory functions of PapB and PapB-like proteins, with an emphasis on their role in controlling the expression of fimbrial adhesins. This review also explores the interplay between PapB and global regulators, positioning PapB as a central player within the fimbrial regulatory network. Understanding these mechanisms not only sheds light on bacterial pathogenesis but also identifies PapB and its homologs as potential targets for inhibiting some types of fimbriae as a targeted strategy to prevent bacterial infections dependent on Pap B-family regulators.

## 2. P Fimbriae in *E. coli*

### 2.1. Structural Architecture and Biogenesis

The biosynthesis of P fimbriae in *E. coli* is governed by the *pap* operon (*papBAHCDJKEFG*), which encodes both structural and regulatory components. The fimbrial structure is primarily composed of the major subunit PapA, while the tip adhesin PapG mediates host receptor binding. The minor subunits PapE, PapF, and PapK form the tip fibrillum, contributing to host specificity (Figure 1). Assembly is orchestrated by the periplasmic chaperone PapD and the outer membrane usher PapC (chaperon-usher system), which coordinate the ordered secretion and polymerization of subunits, ensuring functional fimbrial display on the bacterial surface (Figure 2) [9,13].

### 2.2. Epigenetic and Global Regulatory Control

The expression of P fimbriae in *E. coli* is governed by a finely tuned regulatory system that integrates epigenetic modifications, environmental signals, and global transcriptional control. Central to this system are the divergently transcribed *papB* and *papI* genes, which encode key regulators of the *pap* gene cluster. The regulatory logic of this locus is mediated through phase variation, controlled by the accessibility of promoter regions marked by GATC motifs, which are substrates for DNA adenine methylation by Dam [24,25,27,30]. Leucine-responsive regulatory protein (Lrp) binds to two distinct clusters flanking the promoter, and its interaction is mutually exclusive with Dam-mediated methylation, providing a reversible ON/OFF switch for fimbrial expression [27,30]. In the OFF state, unmethylated GATC sites near the *papBA* promoter are bound by Lrp, which represses *papBA* transcription while simultaneously enhancing *papI* expression. PapI subsequently interacts with Lrp to form a complex that promotes transcriptional activation by displacing Lrp from repressive sites and allowing Dam to methylate the DNA. Methylation of GATC sites diminishes Lrp binding affinity, thereby shifting the operon into the ON state and enabling expression of the fimbrial subunits. This system allows *E. coli* to respond dynamically to changes in the host environment by controlling fimbrial adhesin expression through reversible epigenetic modifications [31,32,33]. PapB acts downstream of PapI and serves a dual role in regulating *PapBA* depending on its intracellular concentration. At low to moderate concentrations, PapB enhances transcription of *papBA* by modulating DNA topology and possibly counteracting Lrp repression. At higher concentrations, however, PapB autoregulates its own expression and represses *papBA* to avoid excessive fimbrial production [23,34,35,36]. The intermediate inactive state of the *pap* gene cluster is marked by low *papBA* transcription and a slight elevation of *papI* expression. In this state, Lrp binds to the unmethylated GATC site near the *papBA* promoter, inhibiting *papBA* transcription while promoting *papI* expression. The binding of Lrp to either the upstream or downstream clusters prevents DNA adenine methyltransferase (Dam) from methylating the GATC site. Methylation by Dam reduces Lrp’s affinity for the DNA, modulating its regulatory role [25,27,32]. Despite this, the binding of Lrp to the GATC site is not entirely abolished by methylation. PapI, in complex with Lrp, binds to an unmethylated GATC site near the *papI* promoter, activating *papBA* transcription and turning the operon ON. Additionally, PapB autoregulates its expression to maintain controlled fimbrial production under varying conditions (Figure 1) [23,27,36].

In addition to the Lrp-Dam-PapI-PapB axis, global regulators such as H-NS contribute to environmental modulation of *pap* expression. Under non-permissive conditions—such as low temperature, high osmolarity, or nutrient-rich conditions—H-NS binds to AT-rich regions within the *pap* locus to silence *papBA* and *papI* expression by forming repressive nucleoprotein filaments [31]. This repression is reversible and is influenced by environmental cues, reinforcing the phase-variable nature of P fimbrial expression and enhancing the pathogen’s ability to adapt to changing host niches [27].

Recent studies also highlight the role of a small RNA papR in regulating phase variation at the *pap* locus during UPEC infection in bladder epithelial cells. Transcriptionally activated by Lrp, papR acts as a post-transcriptional repressor of papI. Deletion of papR enhances bacterial adhesion to kidney and bladder cells, even in the absence of type 1 fimbriae, allowing rapid adaptation to changing host environments [28].

## 3. PapB-like Regulators Controlling Fimbrial Expression and Bacterial Pathogenesis

PapB-like regulators are small transcription factors crucial for controlling the expression of fimbrial adhesins in various bacterial species. These regulators often regulate fimbriae encoding systems through response to environmental and host-specific cues. PapB, the prototype of this family, regulates P fimbriae in *Escherichia coli*, with homologs such as SfaB and FocB controlling S and F1C fimbriae, respectively. Similar regulators, like MrpB in *Proteus mirabilis*, govern mannose-resistant fimbriae production in this species [20,21,22].

Some other *E. coli* fimbrial systems also feature PapB-like regulators, including Pix. The *pix* protein, found in uropathogenic *E. coli* (UPEC), aids in bladder colonization, though its role in pathogenesis remains to be fully elucidated [20]. PlfB is another PapB family protein, although at present it is unknown what specific role it may play in the regulation of PL fimbriae [18].

The phylogenetic analysis of PapB-family regulators reveals evolutionary clustering that reflects both functional conservation and divergence among fimbrial systems in *Escherichia coli* and related enterobacteria. The proteins PapB, SfaB, and PefB form a tightly grouped clade, sharing high sequence identity and short branch lengths, indicating strong evolutionary conservation likely linked to their roles as transcriptional regulators of fimbrial operons. This suggests that even closely related regulators can acquire unique regulatory roles depending on the genetic and ecological context. These patterns support the idea that while a conserved regulatory backbone exists among certain fimbrial systems, others have evolved distinct mechanisms, possibly through horizontal gene transfer or niche-specific adaptation (Figure 3). The presence of PapB-like regulators in these gene clusters suggests a shared regulatory strategy that enables bacteria to fine-tune adhesion, balancing colonization and dispersal in response to dynamic environments.

### 3.1. F1C and SFA Fimbriae and the FocB/SfaB Regulators

F1C fimbriae, encoded by the *foc* operon, facilitate the adherence of uropathogenic *Escherichia coli* (UPEC) to urinary tract epithelial cells. More frequently detected in UPEC strains from cystitis cases (14–38%) than in fecal isolates (7%), the *foc* gene cluster closely resembles the *pap* operon in gene organization, differing only in minor rearrangements [41]. It comprises seven structural genes and two regulatory genes near the promoter region. The transcriptional regulator FocB, a member of the PapB family, shares 81% identity with PapB and complete identity with SfaB, which regulates S fimbriae in neonatal meningitis-associated *E. coli*. FocB exhibits partial similarity to PefB, a fimbrial regulator in *Salmonella enterica*, as well as to PapB and SfaB in *E. coli*. [42,43]. F1C fimbriae mediate binding to glycosphingolipids, such as galactosyl ceramides and globotriosyl ceramides, prevalent in the bladder, ureters, and kidneys, supporting UPEC colonization and persistence [41]. Structurally, the F1C fimbriae are composed of FocA as the major subunit and a tip adhesin complex including FocF, FocG, and FocH. Its biogenesis follows the chaperone-usher pathway, involving the periplasmic chaperone FocC and outer membrane usher FocD. The expression of the *Foc* operon is regulated by SfaC and FocB, homologs of PapI and PapB, respectively [2,42].

FocB specifically functions as a transcriptional regulator that binds to upstream regulatory regions of the *foc* and *pap* operons, modulating their expression through interaction with global regulators such as H-NS and Lrp. As demonstrated by Hultdin et al. [42], FocB has a winged helix-turn-helix (wHTH) domain structurally analogous to PapB and can displace the repressor H-NS from operator DNA, relieving repression and enabling Lrp-mediated activation [42]. This cross-talk enables FocB to not only activate the *Foc* operon but also influence expression of *pap* genes under certain conditions, effectively integrating the regulation of multiple fimbrial systems. Co-regulation by FocB or SfaC provides tight control and environmental responsiveness, ensuring that energetically costly fimbrial systems are expressed only when beneficial to the pathogen [44]. Emerging evidence also suggests a role for F1C fimbriae in biofilm formation, contributing to UPEC adaptability in host environments [41].

S fimbriae are important adhesins associated with extra-intestinal pathogenic *E. coli* that facilitate attachment to sialylated glycoproteins in brain endothelial and renal tissues, contributing to neonatal meningitis and sepsis. Encoded by the *sfa* gene cluster, this ~6.5 kb locus includes at least seven genes, among them the 16.5 kDa fimbrial subunit protein [44]. *Sfa* expression is governed by three promoters—pA, pB, and pC—with pB serving as the primary driver of transcription. While the *sfa* operon’s primary promoter, pA, remains largely inactive under non-inducing conditions, it can produce a 700-base transcript spanning *sfaA*, along with additional 500- and 1400-base transcripts corresponding to *sfaC* and *sfaBA*, respectively [45]. In contrast to the *pap* gene cluster, where PapB and CRP-cAMP facilitate activation, *sfa* expression is positively regulated by SfaB and SfaC. A proposed repressor, SfaR, may inhibit transcription in the absence of these activators [21,46], and *hns* mutations have also been shown to relieve repression and activate *sfaA* expression [46,47].

Previously thought to be governed solely by direct transcriptional control, emerging evidence has shown that *sfa* expression is also modulated by phase variation mechanisms involving the leucine-responsive regulatory protein (Lrp) and DNA adenine methylase (Dam). Lrp binding influences local DNA methylation at GATC sites within the *sfa* promoter region, creating mutually exclusive methylation states that determine whether the operon is in the transcriptionally active (ON) or inactive (OFF) phase. This mechanism parallels the regulation of the *pap* operon and highlights that S fimbriae, like P fimbriae, are subject to epigenetically controlled, reversible phase variation mediated by Lrp and Dam [48]. This regulatory divergence likely reflects an evolutionary adaptation favoring rapid and transient adhesion during systemic infections [25,45].

### 3.2. Pef Fimbriae and the PefB Regulator

Pef fimbriae are fimbrial adhesins that promote *S. enterica* Typhimurium colonization of intestinal epithelium, facilitating resistance to peristalsis and initiating gastrointestinal infection. This adhesion triggers virulence mechanisms leading to inflammation, fluid secretion, and diarrhea. The *pef* operon, comprising genes *pefA*, *B*, *C*, *D*, and *I*, is finely regulated to function under the acidic and fluctuating pH of the intestine [22].

Regulation of *pef* transcription mirrors that of the *pap* system, relying on Dam-mediated methylation of GATC motifs within the promoter region. Lrp acts as a positive regulator, enhancing transcription, while H-NS and RpoS suppress expression under non-inducing conditions. Notably, *pefI*, although homologous to *papI*, functions as a repressor rather than an activator. PefB, a paralog of PapB, plays an important but less-characterized role in the regulatory control of the *pef* operon. Although its function in *Salmonella* remains underexplored, comparative studies suggest that PefB likely contributes to transcriptional regulation of fimbrial genes by interacting with promoter regions or modulating nucleoid-associated silencing proteins [49,50]. In *Escherichia coli*, PefB has been shown to repress *fimB*-mediated recombination, suggesting that it may also influence phase variation or expression of other fimbriae indirectly [44,46].

Its sequence similarity to PapB (~46% identity) and conservation of a helix-turn-helix DNA-binding motif support the likelihood that it participates in fine-tuning the expression of *pef* genes under specific environmental conditions [22]. Moreover, the *pef* operon contains two promoters, *PpefB* and *PpefA*, and is subject to repression by H-NS, Hha, and StpA, especially in the absence of host-inducing cues [51]. While the precise mechanisms by which PefB integrates into this network are not fully delineated, it is likely that PefB acts in coordination with Lrp and Dam methylation to modulate expression in a context-dependent manner. This distinct regulatory arrangement ensures optimal expression of fimbriae for intestinal colonization and pathogenesis [51,52].

### 3.3. Pix Fimbriae and PixB Regulator

Pix fimbriae, part of the π-fimbrial family, are adhesins that contribute to the virulence of uropathogenic *Escherichia coli* (UPEC) by promoting attachment to uroepithelial cells during urinary tract infections. These surface structures enhance bacterial colonization and persistence in the bladder and kidneys, exacerbating infection severity [20].

Encoded by the *pix* operon, which mirrors the genetic organization of the *pap* locus, Pix fimbriae are assembled via the chaperone-usher pathway. Core components include PixA (major pilin), PixG (tip adhesin), PixD (periplasmic chaperone), and PixC (outer membrane usher), with PixB functioning as a regulatory factor. There is, however, no PapI-type protein present in the *pix* regulatory region. Expression of the *pix* operon is controlled by phase variation, enabling reversible ON/OFF switching in response to environmental cues and immune pressures, thus facilitating bacterial adaptability and possible immune evasion during infection [35].

### 3.4. Orf G from Plasmid pMB2: A Non-Fimbrial Member of the PapB Regulatory Family

Orf G, encoded by the conjugative plasmid pMB2, is a plasmid-borne transcriptional regulator identified in *Escherichia coli* transconjugants. Unlike chromosomally encoded PapB-family members typically associated with fimbrial operons, orf G is not linked to any fimbrial structural gene cluster, yet it modulates phenotypes related to surface behavior, particularly auto-aggregation [49]. The gene was initially described by Monárrez and Okeke [49] who demonstrated that its expression significantly altered cell aggregation patterns, suggesting a role in host cell interaction or plasmid dissemination. Structurally, Orf G belongs to the PapB family of transcriptional regulators and shares sequence homology with PapB, SfaB, and FocB, although it shares the highest level of identity with TosR [53]. While Orf G lacks an associated chaperone-usher fimbrial operon, it likely retains a conserved helix-turn-helix (HTH) domain, enabling it to influence gene expression through DNA binding. Its plasmid location and functional divergence suggest that PapB-like regulators have been horizontally transferred and adapted to new regulatory contexts beyond fimbrial control. As such, Orf G represents a unique expansion of the PapB regulatory family, emphasizing the plasticity and regulatory versatility of these small transcription factors in both chromosomal and extrachromosomal environments [44]. Although the downstream targets of Orf G remain to be fully characterized, its influence on bacterial aggregation hints at broader roles in plasmid biology, host colonization, or interbacterial interactions [49].

### 3.5. Additional Regulators Belonging to the PapB Family

Other regulators belonging to the PapB family include FaeA F4 (K88) fimbriae, DaaA of F1845 fimbriae, ClpB of CS31 fimbriae, Fan A and Fan B of K99 fimbriae, AfaA regulating the Afa adhesins, DaaA of F1845 fimbriae and TosR. These other systems will be described in the following sections.

## 4. FaeA: Regulation of F4 (K88) Fimbriae in Enterotoxigenic *E. coli*

F4 (K88) fimbriae, a key virulence factor of enterotoxigenic *Escherichia coli* (ETEC), play a crucial role in the colonization of the porcine small intestine, leading to diarrhea in neonatal and postweaning pigs. The ability of ETEC to adhere to the intestinal epithelium via F4 fimbriae facilitates bacterial persistence and subsequent secretion of enterotoxins, which drive disease pathology [16]. The *fae* operon consists of nine genes involved in fimbrial biosynthesis, structure, and regulation. The operon includes genes encoding structural subunits (faeG, faeC, faeF, faeH, faeI, and faeJ), as well as the periplasmic chaperone FaeE and the outer membrane usher FaeD. The operon also contains regulatory proteins, including FaeA and FaeB, which influence transcriptional control. Unlike many fimbrial systems where adhesins are confined to the tip structure, the binding properties of F4 fimbriae are distributed along the entire shaft of the fiber, allowing for enhanced interaction with host receptors. Transcription of the *fae* operon is tightly regulated by the operon-specific regulator FaeA and the global regulator Lrp, ensuring controlled expression under appropriate environmental conditions. Although FaeB is encoded within the operon, its function remains unclear, unlike its counterpart PapB, which serves as a positive regulator in the *pap* gene cluster [15].

## 5. Functional and Regulatory Comparison of DaaA and PapB

The *daa* operon of diffusely adherent *Escherichia coli* (DAEC) encodes the F1845 fimbriae, a virulence factor critical for colonization of the intestinal epithelium. Among the operon’s gene products, DaaA is a small regulatory protein that shares structural similarity with the PapB family of transcriptional regulators, including PapB, SfaB, and FocB [31]. These proteins are known for their role in controlling the expression of fimbrial adhesins through direct interaction with upstream regulatory regions and global nucleoid-associated factors. DaaA, however, while structurally related to PapB, exhibits important differences in regulatory activity and functional scope [1,46].

Sequence analyses reveal that DaaA contains a conserved helix-turn-helix (HTH) DNA-binding motif similar to that of PapB, yet it lacks critical residues in the carboxy-terminal domain required for PapB’s broader regulatory function. Functional assays show that while PapB represses type 1 fimbriae expression by inhibiting *fimB*-mediated recombination at the fim switch (*fimS*), DaaA does not influence the orientation of *fimS*, nor does it affect *fimA* transcription. Substitution of PapB’s C-terminal residues with those from DaaA significantly diminishes its ability to repress type 1 fimbriae, suggesting that the unique sequence composition of DaaA confines its regulatory influence to its native operon rather than allowing cross-regulation between fimbrial systems [10,31,54].

Regulatory mechanisms within the *daa* operon also differ markedly from those of the *pap* operon. While PapB integrates environmental signals via Lrp, Dam methylation, and H-NS to coordinate phase variation in P fimbriae, DaaA’s regulation operates primarily at the transcriptional and post-transcriptional levels. Transcription of *daaE*, the major fimbrial subunit gene, is repressed under non-inducing conditions, such as low temperature or high osmolarity by the nucleoid structuring protein H-NS [31]. This repression is relieved in *hns* mutants, indicating an environmental responsiveness analogous to other fimbrial operons. Interestingly, expression of *daaE* is also regulated post-transcriptionally by the upstream open reading frame *daaP*. Translation of *daaP* is required for proper mRNA processing of the downstream *daaE*, a regulatory feature not observed in the *pap* system [55].

In contrast to PapB, which is deeply embedded in a network of cross-operon regulatory influences—such as repressing type 1 fimbriae and responding to the CRP-cAMP pathway—DaaA appears to function as a more isolated, operon-specific regulator. This divergence may reflect evolutionary adaptation to niche-specific regulatory demands of DAEC in the intestinal tract. Taken together, while DaaA and PapB belong to the same structural family of fimbrial regulators, their differing roles in cross-talk, operon regulation, and environmental responsiveness underscore the functional diversification of fimbrial control mechanisms in *E. coli* pathotypes [56].

## 6. ClpB of the CS31A Fimbriae

The CS31A fimbriae, encoded by the *clp* operon in certain pathogenic *E. coli* strains, play a key role in adherence to host tissues during colonization of the intestinal tract, particularly in calves. Central to the regulation of this system is ClpB, a small transcriptional regulator structurally related to PapB, the well-characterized regulator of P fimbriae in uropathogenic *E. coli*. While both ClpB and PapB belong to the same helix-turn-helix (HTH) protein family, they differ markedly in their regulatory roles and scope of influence [57]. ClpB and PapB share conserved structural elements, especially in their DNA-binding domains, suggesting a common evolutionary origin. However, functional divergence is apparent. PapB not only activates the expression of its own *pap* operon but also exerts trans-repressive effects on the type 1 fimbrial operon by interfering with *fimB*-dependent recombination and promoting *fimE*-mediated switching to the “OFF” orientation of the *fimS* phase switch. In contrast, ClpB lacks the ability to influence type 1 fimbriae expression, indicating the absence of regulatory cross-talk. Experimental evidence confirms that ClpB does not repress *fimB* or modulate fimbrial phase variation, underscoring its specificity to the CS31A system [57].

The regulation of the *clp* operon is driven by both local and global transcriptional factors, primarily the leucine-responsive regulatory protein (Lrp). Lrp binds to upstream regions of the *clp* promoter and represses transcription under nutrient-rich conditions. This repression is alleviated by branched-chain amino acids, especially leucine and alanine, indicating a fine-tuned regulatory mechanism responsive to host nutrient availability [57]. Notably, the *clp* operon is not under the control of Dam methylation or known phase variation systems. Furthermore, current evidence suggests that ClpB does not engage in cross-regulation of other fimbrial systems such as type 1 or P fimbriae, distinguishing it from regulators like PapB, which orchestrate broader regulatory effects influencing the adhesin expression hierarchy. The absence of cross-talk implies that ClpB acts exclusively within its native regulatory circuit, maintaining tight control over CS31A expression without interfering with the transcriptional programs of other adhesin operons [54,57].

The transcriptional regulation of the *clp* operon also diverges mechanistically from that of *pap*. PapB integrates multiple regulatory inputs, including Lrp, Dam methylation, and environmental signals, to coordinate P fimbriae expression. In contrast, ClpB operates within a more streamlined network, primarily responding to nutrient cues through Lrp. These features highlight a more specialized and environmentally responsive regulation of CS31A fimbriae, in contrast to the complex, multilayered control seen in P fimbriae [57].

## 7. Regulatory Functions of FanA and FanB of the K99 Fimbrial Operon

The K99 fimbriae (also known as F5 fimbriae) are important virulence factors expressed by enterotoxigenic *E. coli* (ETEC), especially in neonatal and young animals, where they mediate adherence to the small intestinal epithelium. The biogenesis and expression of K99 fimbriae are controlled by the *fan* operon, which includes several genes (*fanA* through *fanH*) encoding structural subunits, chaperone-usher assembly machinery, and two regulatory proteins, FanA and FanB. While PapB—best studied in the context of the *pap* operon encoding P fimbriae in uropathogenic *E. coli* (UPEC)—has been shown to participate in broader fimbrial cross-regulation and phase variation, FanA and FanB appear to represent a more localized and specialized form of transcriptional control [58,59].

FanA and FanB are small basic proteins encoded at the 5′ end of the *fan* operon and expressed from a common promoter upstream of *fanA*. Structurally, they share similarities with PapB, particularly in predicted DNA-binding domains, suggesting a conserved mechanism of interaction with target promoters. However, their regulatory functions differ significantly. While PapB is capable of repressing the expression of type 1 fimbriae, there is no evidence that FanA or FanB participates in such cross-talk with heterologous fimbrial systems. Their role appears to be confined to the activation of downstream genes within the *fan* operon, facilitating the assembly and surface expression of K99 fimbriae [59].

The transcriptional regulation of the *fan* operon is responsive to global regulatory factors such as the Lrp and CRP. Lrp acts as a transcriptional activator under nutrient-limited conditions, enhancing the expression of *fanA* and *fanB*, while CRP mediates catabolite repression, integrating environmental nutrient signals. This regulatory logic allows ETEC to express K99 fimbriae in the nutrient-limited conditions typical of the small intestine, where colonization occurs. Unlike the *pap* operon, the *fan* operon is not phase-variable, nor is it subject to Dam methylation-dependent silencing, suggesting a simpler and more targeted expression strategy tailored to a defined ecological niche [58].

## 8. Comparative Analysis of AfaA and PapB

The Afa/Dr family of adhesins represents a class of afimbrial adhesins expressed by diffusely adherent *E. coli* (DAEC) and certain uropathogenic *E. coli* (UPEC) strains. These adhesins are implicated in urinary tract infections and chronic diarrhea, particularly in children and immunocompromised patients [60,61]. Central to the expression of these adhesins is the *afa* operon, which encodes several proteins involved in assembly, transport, and regulation of the adhesin system. Among these, AfaA serves as a key regulatory protein and shares structural and functional similarities with PapB, the prototypical transcriptional regulator of the *pap* operon encoding P fimbriae. A comparison of AfaA and PapB reveals both evolutionary homology and significant functional divergence in their mechanisms of regulation, operon organization, and roles in pathogenesis [62].

The *afa* operon typically comprises five genes: *afaA*, *afaB*, *afaC*, *afaD*, and *afaE*. AfaA is located at the 5′ end of the operon and encodes a small cytoplasmic protein with predicted helix-turn-helix motifs similar to those found in PapB. Like PapB, AfaA is believed to function as a DNA-binding protein that modulates the transcription of downstream genes required for adhesin production. However, while PapB not only activates its operon but also participates in trans-regulation, notably repressing type 1 fimbriae by influencing the *fim* switch, AfaA appears to act exclusively in a cis-regulatory manner within the *afa* operon. To date, there is no evidence that AfaA influences the expression of other fimbrial or afimbrial systems, nor does it appear to play a role in phase variation mechanisms that typify PapB function [1,63,64].

The regulation of the *afa* operon is also shaped by interactions with global regulators and environmental cues. For instance, host-derived factors such as temperature and osmolarity can affect the transcriptional activity of the *afa* promoter. Although the precise molecular interactions remain less well-characterized than those of PapB, it is clear that AfaA operates within a streamlined, operon-specific regulatory architecture, optimized for the persistent, low-level expression of Afa/Dr adhesins in host mucosal environments. By contrast, PapB functions within a broader regulatory network, coordinating fimbrial expression hierarchies in response to environmental inputs through interactions with Dam methylation, Lrp, H-NS, and recombinase systems. This allows PapB to exert control over both phase-variable and non-phase-variable operons, tailoring fimbrial expression profiles across stages of infection and tissue niches [1,60,65].

## 9. PapB in Relation to Other Fimbrial Regulators

FimB and FimE, the key regulators of type 1 fimbrial phase variation in *Escherichia coli*, have a distinct mechanism of action when compared to PapB, the master regulator of the *pap* (P fimbriae) fimbrial gene cluster. FimB and FimE are site-specific recombinases that control expression via an invertible DNA element, with FimB mediating bidirectional switching, while FimE favors the OFF orientation [66]. Notably, the *fim* and *pap* gene clusters are genetically distinct but evolutionarily and mechanistically analogous. PapB itself is a small DNA-binding protein that acts as both a positive and negative regulator by modulating promoter accessibility within the *pap* gene cluster [35,67,68,69,70]. Sequence homology and regulatory behavior suggest that fimbrial regulators, such as TosR, homologous to PapB, may have co-evolved to govern fimbrial and non-fimbrial adhesin expression across diverse pathogenic *E. coli* strains. Studies on the Tos operon in uropathogenic *E. coli* further support the evolutionary plasticity of PapB-like regulators. TosR, for example, represses its own operon while simultaneously interacting with promoters of other adhesins like Pap, suggesting a broad-reaching regulatory network among fimbrial and non-fimbrial loci. These findings support the concept that non-PapB regulators, such as FimB, while lacking identical sequences, fulfill overlapping roles in coordinating adhesin expression in response to environmental or host-derived cues [70].

## 10. Cross-Talk Between Fim Regulation and the PapB Family Regulators

Cross-regulatory mechanisms exist between different fimbrial systems in *E. coli*, enabling precise temporal control of surface structures to adapt to host environments. In uropathogenic strains, expression of type 1 fimbriae inversely correlates with the expression of P fimbriae, a phenomenon termed fimbrial cross-talk [71]. This mutual exclusivity is mediated by regulators such as PapB, Lrp, and H-NS, which directly or indirectly influence the orientation of the *fim* switch (*fimS*). The interplay ensures energy-efficient and host-context-appropriate expression of adhesins, allowing bacteria to modulate adherence profiles during different stages of infection. Notably, PapB itself can repress type 1 fimbrial expression by interfering with the transcriptional activity at the *fim* switch region. Engström and Mobley demonstrated that PapB homologs like TosR bind AT-rich promoter regions and downregulate P fimbriae expression, which in turn may permit type 1 fimbrial activation depending on the chromatin context [70]. Such hierarchical regulation suggests a broader adhesin regulatory axis, wherein the expression of one adhesin system dynamically suppresses another.

This regulation is reinforced by environmental sensing systems and nucleoid-associated proteins like H-NS and Lrp, which participate in fine-tuning fimbrial gene accessibility [27].

Comparative studies on regulatory cistrons across fimbrial systems have shown that PapB and its homologs—SfaB, FocB, PefB, and TosR—share a conserved helix-turn-helix DNA-binding motif and can functionally substitute for one another to a certain extent [72]. These regulators typically bind to upstream AT-rich regions and modulate transcription either by antagonizing global repressors like H-NS or by recruiting activators such as Lrp. For example, SfaB and FocB—although sometimes encoded in different contexts—have both been shown to influence the expression of unrelated fimbrial operons and even affect the *fim* switch orientation indirectly. This modularity indicates a shared evolutionary strategy among ExPEC strains, allowing for regulatory flexibility during infection [72,73].

Furthermore, PapB-family proteins do not limit their influence to adhesin expression alone. TosR, for instance, has been implicated in regulating motility by modulating the *flhDC* flagellar master operon, thereby linking fimbrial control to flagellar function. This coupling enables *E. coli* to shift between sessile (adherent) and motile states depending on environmental signals, infection phase, or anatomical niche. Such integrated regulation enhances bacterial adaptability by coordinating surface structure deployment with metabolic and motility programs, ensuring successful colonization and immune evasion. These insights underscore that fimbrial regulation in *E. coli* is not isolated, but rather interconnected with broader transcriptional networks and pathogenesis strategies [74].

## 11. Discussion and Conclusions

The comparative analysis of fimbrial gene clusters such as *pap*, *pix*, *sfa*, *foc*, and *pef* highlights a conserved modular architecture, with regulatory genes like *papB* playing pivotal roles in modulating downstream fimbrial expression. The striking similarities in operon structure and regulatory sequences, especially the conserved positioning of genes such as *papB* and the involvement of transcriptional regulators like H-NS, Lrp, and CRP, underscore the functional significance of precise regulatory control in pathogenic contexts. These systems are tightly regulated to ensure optimal expression during host colonization, preventing unnecessary energy expenditure and immune detection. The detailed regulatory circuit of the *Foc* operon, which mirrors that of *pap*, demonstrates how bacterial pathogens integrate environmental cues to fine-tune fimbrial gene expression.

Importantly, a broader inclusion of PapB family homologs—including SfaB, FocB, PefB, PixB, and TosR—reveals that this family of regulators extends well beyond the canonical Pap system and serves as a central regulatory network in multiple fimbrial operons. These proteins often share helix-turn-helix motifs and bind to AT-rich regions near promoter elements, though their regulatory capacity and interaction networks may differ significantly. For example, while PapB and SfaB show similar repression of type 1 fimbriae by modulating the fim switch, FocB may act in a more confined regulatory context, and PefB (though homologous) has been shown to repress rather than activate transcription. Furthermore, TosR, a PapB-like transcriptional regulator encoded on pathogenicity islands in UPEC, has been implicated in both repression of P fimbriae and activation of other loci such as iron uptake systems and motility genes, indicating a broader role in virulence regulation. These functional differences support the idea that PapB homologs have evolved diverse regulatory strategies despite their conserved structural domains.

From a therapeutic standpoint, targeting fimbrial regulation remains a promising anti-virulence strategy. Inhibitors that block PapB-family regulators or mimic competitive global regulators like Lrp could disrupt transcriptional activation of fimbrial genes, effectively silencing bacterial adhesion without applying selective pressure typically associated with antibiotics. This could be particularly valuable in managing recurrent urinary tract infections (UTIs), where type 1 and P fimbriae play critical roles in bladder colonization and biofilm formation. Further, identifying cues or regulatory signals that can increase expression of PapB family regulators, could also lead to effective strategies to inhibit expression of other fimbriae such as type 1 fimbriae, that play a critical role in pathogenesis of urinary tract infections caused by UPEC.

Moreover, small molecules that interfere with protein-DNA interactions at conserved regulatory sites, such as Site 1 and Site 2 found in *pap* and *foc* promoters, could have broad-spectrum utility against multiple fimbrial operons that rely on structurally similar regulatory architectures.

From an evolutionary perspective, the recurrence of PapB-like regulatory modules across diverse fimbrial systems supports the idea of horizontal gene transfer and strong selective pressures to maintain such regulators in uropathogenic *E. coli* lineages. Phylogenetic comparisons show that PapB, SfaB, and FocB cluster closely together, suggesting recent divergence from a common ancestor, whereas more distantly related homologs like PefB branch separately, reflecting possible specialization or regulatory rewiring. The inclusion of TosR in this comparison further supports the expansion of the PapB regulatory family into horizontally acquired and chromosomally encoded loci that control a wider array of virulence functions. The preservation of gene order and the use of similar promoter elements across operons such as *pap*, *sfa*, *foc*, and *pef* imply convergent evolution toward a successful regulatory template for adhesion control. Understanding these evolutionary and mechanistic relationships enhances our grasp of fimbrial plasticity and provides a strong foundation for developing molecular inhibitors targeting PapB-family regulators to prevent or limit infections dependent on adhesin expression controlled by PapB-family protein-mediated regulatory mechanisms.

## Figures and Tables

**Figure 1 microorganisms-13-01939-f001:**
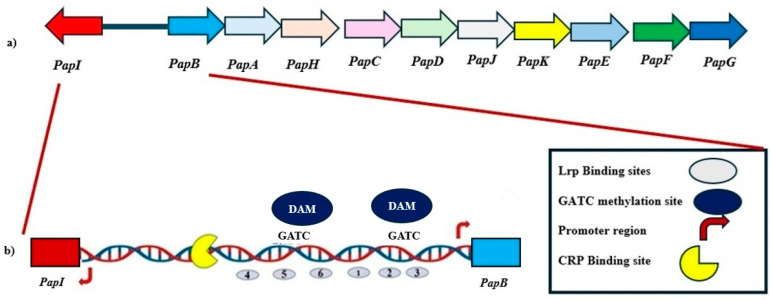
Genetic Organization and Regulatory Architecture of the *pap* Operon. (**a**) The *pap* gene cluster in *Escherichia coli* encodes multiple genes responsible for the assembly and regulation of P fimbriae. The gene cluster is divergently transcribed, with the regulatory genes *papI* and *papB* located upstream of structural genes (*papA* through *papG*). (**b**) The intergenic region between *papI* and *papB* contains multiple regulatory elements that coordinate phase-variable expression of P fimbriae. This includes Leucine-responsive Regulatory Protein (Lrp) binding sites (gray ovals), GATC motifs (blue circles) that serve as methylation targets for DNA adenine methyltransferase (Dam), and promoter regions for *papI* and *papB*. Methylation of specific GATC sites influences Lrp binding, thereby modulating promoter accessibility and gene expression. A cAMP receptor protein (CRP) binding site (yellow crescent) further integrates environmental signals into the regulatory output. This configuration supports a reversible ON-OFF switch controlled by epigenetic modifications and regulatory proteins, enabling dynamic adaptation of fimbrial expression during infection.

**Figure 2 microorganisms-13-01939-f002:**
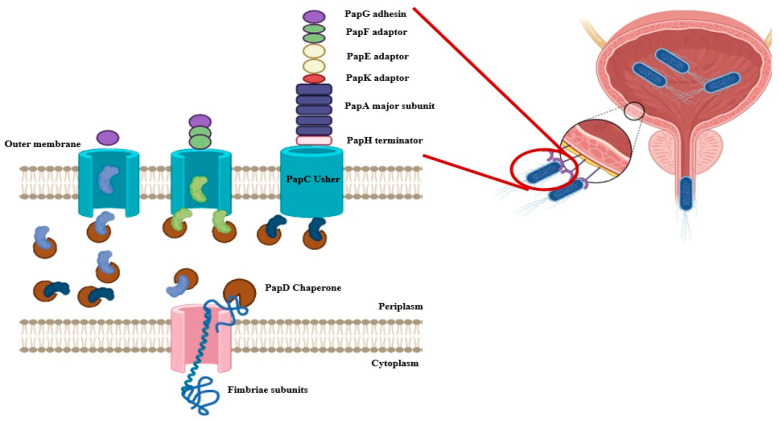
Biogenesis and Assembly of P Fimbriae in *Escherichia coli*. The figure illustrates the chaperone-usher pathway responsible for the assembly and surface localization of P fimbriae in *E. coli*. Fimbrial subunits, including PapG (adhesin tip, purple), PapF (green), PapE (minor fibrillin, yellow), PapK (adaptor, red), PapA (major rod subunit, blue), and PapH (terminator subunit, pink), are translocated into the periplasm through the inner membrane (IM) via the Sec translocon. In the periplasm, PapD acts as a chaperone to stabilize these subunits and prevent premature aggregation. PapD-subunit complexes are then delivered to the outer membrane (OM) usher protein PapC, which facilitates the ordered assembly and secretion of fimbrial subunits onto the bacterial surface. PapC enables the polymerization of PapA subunits into the rod structure, capping the assembly with the tip adhesin PapG and anchoring it with PapH. This highly coordinated process enables the formation of functional P fimbriae that mediate adherence to host epithelial cells, a key step in uropathogenic *E. coli* colonization (Created by Biorender).

**Figure 3 microorganisms-13-01939-f003:**
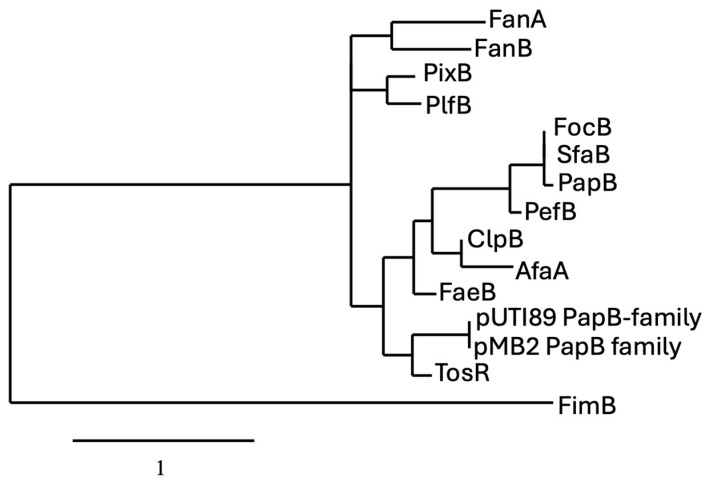
Phylogram of proteins belonging to the PapB regulator family. For comparison of the protein sequences, entries were obtained from either the NCBI (https://www.ncbi.nlm.nih.gov) or the Universal Protein Resource (UniProt) (www.uniprot.org) website. Phylogenetic analyses of protein sequences were conducted with the platform at Phylogeny (http://www.phylogeny.fr) using the default (“one click”) parameters [37]. Analyses consisted of multiple-sequence alignment with MUSCLE [38], alignment curation with GBlocks [39], maximum likelihood phylogeny analysis using PhyML 3.0 [40], and TreeDyn for generation and editing of trees (https://www.phylogeny.fr/one_task.cgi?task_type=treedyn). Specific parameters are described at the Phylogeny.fr website platform. Sequences included in the tree are from Uniprot.org. unless indicated otherwise and include from top to bottom: FanA (P07104); FanB (P07105); PixB (Q83XC7); PlfB (NCBI: AKG46875.1); FocB (Q93K76); SfaB (Q93K76); PapB (P0474); PefB (H9L498); ClpB (Q47101); AfaA (P53515); FaeB (Q47205); pUTI89 PapB-family (Q1R1Z3); pBM2 PapB-family (NCBI: WP_001322642); TosR (A0AAD2S3G5); FimB (P0ADH5), which was used as an outlier for the tree, (P0ADH5).

## Data Availability

No new data were created or analyzed in this study.

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
