# Peer review of "PapB Family Regulators as Master Switches of Fimbrial Expression"

_microorganisms, 2025, doi:10.3390/microorganisms13081939_

Round 1
Reviewer 1 Report
Comments and Suggestions for Authors
The review paper by Akrami sumarized the discovery and funnctions of PapB family proteins in bacteria. The topic is interesting and somewhat significant in various aspects for understanding bacterial virulence and growth. This reviewer has no major issues for this paper and suggests some minor comments for authors to improve the paper.
1) The font size in most figure is too small and thus needs reconsideration. The color also not appropriate in Figure 1.
2) The overall logic and flow of the review structure is bit hard to follow. Some connections seem lost between each paragraph. This reviewer suggests reorignize the paper with some sections and subsections to increase the readibility of the paper.
3) As for the discussion and perpectives, the authors need expand this section with more constructive comments and solutions for the further directions of PapB family regulators.
4) Some proofreading is required to avoid any overstatment and mistake/typos berfore resubmissions.
Author Response
Thank you for the review of our manuscript. We appreciate the comments and critiques that you have raised that have allowed us to improve the text and clarity of the manuscript. The changes are indicated in red in a marked copy of the manuscript.
Here are the specific responses to your comments:
1) The font size in most figure is too small and thus needs reconsideration. The color also not appropriate in Figure 1.
We have altered Fig. 1 by increasing the font size of the letters and placed the names outside of the arrows or blocks in Fig. 1a and Fig.1b.
2) The overall logic and flow of the review structure is bit hard to follow. Some connections seem lost between each paragraph. This reviewer suggests reorignize the paper with some sections and subsections to increase the readibility of the paper.
Thank you for this comment. We have included additional subtitles and re-organized sections in order to better structure the logic of the sections.
3) As for the discussion and perpectives, the authors need expand this section with more constructive comments and solutions for the further directions of PapB family regulators.
We do not want to overstate or elaborate too much on the possibilities and opportunities that further understanding of the PapB family may provide. We did, however include some additional text concerning how understanding regulation of these genes could be of practical use through potential treatment/prevention of UTIs caused by UPEC. (Lines 572-575 of the revised text in marked version).
4) Some proofreading is required to avoid any overstatment and mistake/typos berfore resubmissions.
We have proofread the document and have made some minor changes to the text.
We have included a marked copy of the manuscript indicating changes to the manuscript text or section orders. Changes are highlighted.
Reviewer 2 Report
Comments and Suggestions for Authors
In the present review, the authors have explored the structural organization, biogenesis, and multi-tiered regulatory control of P fimbriae, with emphasis on PapB and homologous regulatory proteins such as SfaB, FocB, PixB and PefB. In addition, they have performed comparative genomics and phylogenetic analyses, which revealed that regulators belonging to the PapB family are evolutionarily conserved across π-fimbrial systems and also regulate other types of fimbriae. This is a well-organised and written review, which provides new insights and a basis for future studies on the the genomics of fibrial structure and function. The introduction describes accurately the background of the study, the results are clearly presented and the discussion and conclusions are in accordance with the results. The references are updated and properly cited. The manuscript is accaptable for pubication in its current form.
Author Response
Comments for Reviewer 2.
Thank you for your appreciation of our review manuscript. We appreciate your comments and are glad you found the work of interest and worthy of publication.